# Chromosome-level genome assemblies reveal genome evolution of an invasive plant *Phragmites australis*
Cui Wang [1,2], Lele Liu [1], Meiqi Yin[1], Bingbing Liu [3], Yiming Wu[1], Franziska Eller[4], Yingqi Gao [3], Hans Brix[4], Tong Wang[5], Weihua Guo[1,7] ✉ & Jarkko Salojärvi [2,6,7] ✉

Biological invasions pose a significant threat to ecosystems, disrupting local biodiversity and ecosystem functions. The genomic underpinnings of invasiveness, however, are still largely unknown, making it difficult to predict and manage invasive species effectively. The common reed (*Phragmites australis*) is a dominant grass species in wetland ecosystems and has become particularly invasive when transferred from Europe to North America. Here, we present a high-quality gap-free, telomere-to-telomere genome assembly of *Phragmites australis* consisting of 24 pseudochromosomes and a B chromosome. Fully phased subgenomes demonstrated considerable subgenome dominance and revealed the divergence of diploid progenitors approximately 30.9 million years ago. Comparative genomics using chromosome-level scaffolds for three other lineages and a previously published draft genome assembly of an invasive lineage revealed that gene family expansions in the form of tandem duplications may have contributed to the invasiveness of the lineage. This study sheds light on the genome evolution of Arundinoideae grasses and suggests that genetic drivers, such as gene family expansions and tandem duplications, may underly the processes of biological invasion in plants. These findings provide a crucial step toward understanding and managing the genetic basis of invasiveness in plant species.

Invasive species introduced from exotic regions pose a threat to indigenous ecosystems, and as one of the major drivers of biodiversity loss they can lead to ecological disasters and significant economic losses[1–3]. The Anthropocene has seen an accelerated spread of invasive species, mediated by the ever-increasing global trade over the last centuries[4]. According to a recent estimate, there are currently more than 3500 harmful invasive alien species and the global economic cost due to alien introductions exceeded 423 billion USD in 2019; considerably more than, for example, the global rice market size, estimated at 287 billion USD in 2021[5]. How and why some species become invasive while others undergoing similar introductions do not is still a conundrum. Certain predisposing phenotypic and physiological traits can be identified, such as short generation time, high growth rate, and high fitness[6–8], but from the evolutionary perspective the genome restructurings that lead to invasiveness should be further investigated. A recent genomics study on *Ambrosia artemisifolia* (common ragweed) suggested the escape of

microbial pathogens, together with introgression from a closely related species to have played a role in its invasiveness[9]. Besides introgression, other hypotheses include the accumulation of transposable elements that may bring in extra genome structural variations and genetic diversity, thus contributing to the fast adaptation to new environments observed in invasive species[10]. Another factor commonly suggested to contribute to the invasiveness is polyploidy[11], since a higher level of genomic plasticity in polyploids could allow a wider range of habitats and more resilience[12–14].

Common reed (*Phragmites australis*; Fig. 1a) is a dominant grass species in wetland ecosystems which occupies large coastal areas all around the globe. Apart from its well-known role in water purification and carbon fixation, *P. australis* is also known for its uses in traditional Chinese medicine[15]. As a species with a wide global distribution, the common reed exhibits a high level of intraspecific variation in its functional traits, such as shoot height, shoot density, and flowering occurrence. Genetically, it has

[1]Key Laboratory of Ecological Prewarning, Protection and Restoration of Bohai Sea, Ministry of Natural Resources, School of Life Sciences, Shandong University, Qingdao, PR China. [2]Organismal and Evolutionary Biology Research Programme, Faculty of Biological and Environmental Sciences, University of Helsinki, Helsinki, Finland. [3]Institute of Loess Plateau, Shanxi University, Taiyuan, China. [4]Department of Biology, Aarhus University, Aarhus, Denmark. [5]College of Landscape Architecture and Forestry, Qingdao Agricultural University, Qingdao, China. [6]School of Biological Sciences, Nanyang Technological University, Singapore, Singapore. [7]These authors jointly supervised this work: Weihua Guo, Jarkko Salojärvi. ✉e-mail: whguo@sdu.edu.cn; jarkko@ntu.edu.sg

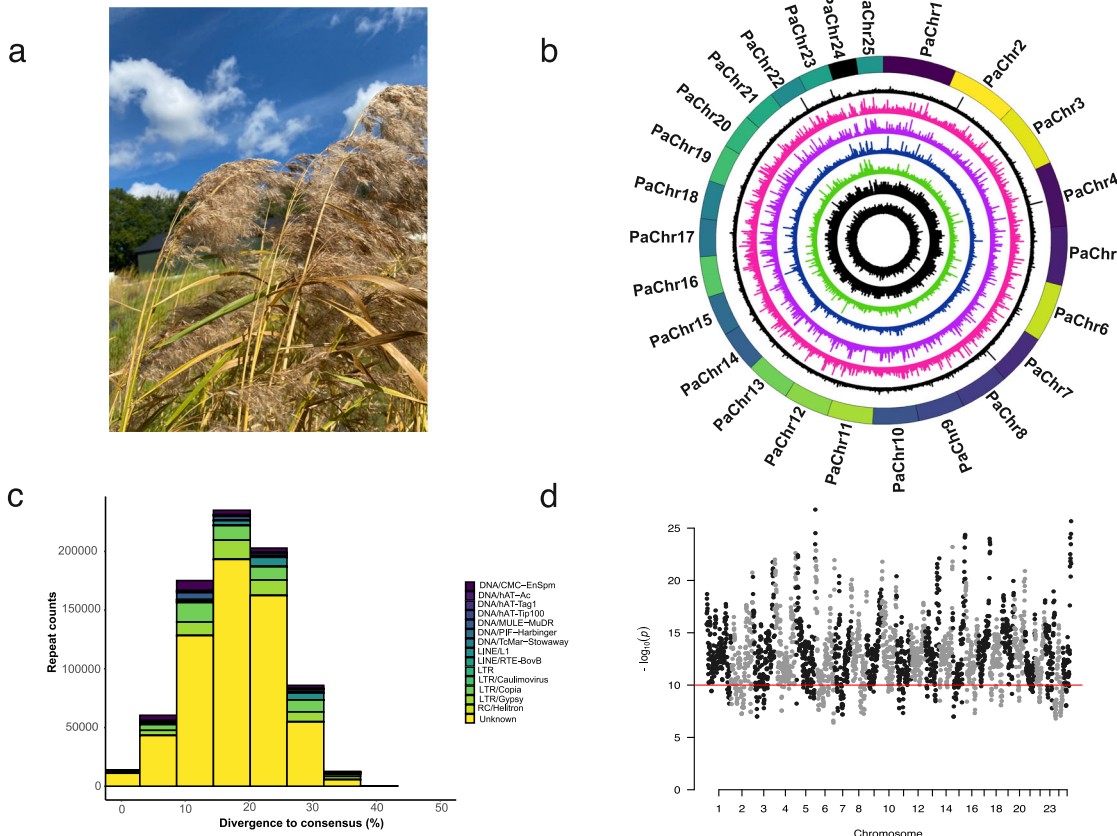

**Fig. 1 | Genome features of the chromosome-level genome assembly of *Phragmites australis*.** Read coverage distribution, repeat content, and LTR assembly index. **a** Photograph of *Phragmites australis* at a common garden in Denmark (Photo: C. Wang). **b** Circos plot of the 25 pseudo-chromosomes in *P. australis*. The outermost circle indicates the length of each chromosome. The histogram tracks, arranged from the outer to the inner circle represent the read coverages for CN, USnat (Y17), USland (Y7), Med (Y21), and EU invasive lineages estimated from mapping illumina short reads against the reference genome. The last two tracks illustrate the gene and repeat contents, respectively. Visualization was conducted from an analysis using a sliding window of 10,000 bp. **c** A stacked bar plot of the different repeat types, split according to their divergence from the consensus repeat sequence (*x*-axis). The major peak shows that a substantial proportion of unknown repeats have similar divergence times, potentially originating from one single expansion event. Expansion events of LTR and LINE repeats may have occurred at different time points. **d** LTR Assembly Index (LAI, *y*-axis) for the 25 chromosomes of the reference genome.

been classified into seven phylogeographic lineages based on chloroplast and nuclear genes. These lineages encompass the indigenous populations in North America, the Mediterranean region, Europe, Northern China, Southern China and Australia, South Africa, as well as the US land type located at the Gulf of Mexico[16]. Common reed is also a super colonizer that shows high adaptability to various environmental conditions[17–19]. Although there are distinct native populations in Europe and North America, the European lineage has become highly invasive in North America after its introduction to the continent approximately 200 years ago, having expanded its range from an initially limited area to span almost the entire United States. During this process, the invasive European lineage has been slowly replacing the native North American populations. Interestingly, while the European lineage is considered invasive in the United States[20], it does not show these characteristics in Europe. In comparison to the native populations in North America, the invasive populations appear to have greener stems, more hairy ligules, and denser seed distribution on the inflorescence[21]. Several factors have been suggested to contribute towards the evolution of invasiveness in *P. australis*, including extensive migration, favorable new environments, asexual propagation, and specific genome evolution[22]. Also, hybridization has been suggested, as introgression between different *Phragmites* species and lineages are often observed, contributing significantly to species diversity[23,24]. From the genomic perspective, the invasive population has been found to possess a larger genome size[25] than its ancestral populations. Increased genome size could result from (segmental) genome duplications, introgression, hybridization, or accumulation of transposable elements, which may have led to genomic changes resulting in the evolution of an invasive lineage. A draft genome of the invasive lineage was recently published[26], but the lack of a chromosome-level assembly and the absence of an assembly from a non-invasive lineage hindered the identification of the specific genome adaptations.

Similar to rice (*Oryza sativa*), the basal chromosome count in genus *Phragmites* is n = 12. In *P. australis*, the euploid chromosome count varies, ranging from 3x to 12x[27]. Tetraploids and octoploids are most common in nature, but mixoploid and aneuploid lineages are also encountered[27,28]. Possibly because of this, also a great extent of intraspecific variation has been documented in terms of morphological traits and molecular markers, and at least three subspecies have been suggested: *P. australis ssp. australis*, *P. australis ssp. altissimus*, and *P. australis ssp. americanus*[29,30]. The versatile ploidy levels in this species allows an opportunity to investigate the potential role of polyploidization in invasiveness.

Here we present a high quality chromosome-level assembly of *P. australis* from PacBio HiFi sequencing. The high-quality reference was used to scaffold four individuals, representing global variation of the species, into chromosome level. The set of genome assemblies was then used to study the genome evolution and demographic histories in *P. australis* lineages to understand the genetic drivers underlying its invasiveness. More specifically, we tested two hypotheses on invasiveness in common reed. First, we evaluated whether the increased genome content in the invasive lineage was involved in proliferation. Second, we tested whether asexual reproduction has facilitated the population growth of common reed and increased

**Table 1 | Genome assembly statistics for the *P. australis* accessions analyzed in this study**

| *P. australis* lineage | CN | Y17 (USnat) | Y21 (Med) | Y7 (USland) | US invasive (Oh et al.) rev* | US invasive (Oh et al.) |
|---|---|---|---|---|---|---|
| Assembled genome size (bp) | 920,401,352 | 922,056,673 | 814,995,062 | 1,217,674,372 | 866,025,734 | 1,139,927,050 |
| Largest contig size (bp) | 55,032,922 | 608,995 | 506,639 | 421,189 | 3,219,705 | 3,219,705 |
| Smallest contig size (bp) | 4080 | 150 | 150 | 150 | 3741 | 1771 |
| N50 contig length (bp) | 34,219,654 | 78,980 | 52,722 | 35,489 | 279,365 | 194,574 |
| Total number of contigs | 1313 | 19,648 | 27,190 | 58,654 | 5335 | 13,411 |
| Sequencing platform | PacBio HiFi + HiC | Illumina WGS | Illumina WGS | Illumina WGS | PacBio RSII | PacBio RSII |
| Percentage of Ns (%) | 0 | 4.98 | 5.34 | 3.7 | 2.09 | - |
| GC (%) | 43.94 | 44.15 | 43.76 | 43.62 | 44.03 | - |
| Number of protein-coding gene loci | 42,498 | 40,688 | 40,969 | 40,124 | 46,625 | 64,857 |
| BUSCO genome (poales_odb10) | | | | | | |
| Complete gene models | 98.9 | 98.1 | 97.4 | 92.5 | 95.9 | 97.2 |
| Single-copy | 47.2 | 48.8 | 51.5 | 58.0 | 55.6 | 41.9 |
| Duplicated | 51.7 | 49.3 | 45.9 | 34.5 | 40.3 | 55.3 |
| Fragmented | 0.1 | 0.6 | 0.9 | 1.9 | 1.3 | 0.9 |
| Missing | 1.0 | 1.3 | 1.7 | 5.6 | 2.8 | 1.9 |
| Busco annotation (poales_odb10) | | | | | | |
| Complete gene models | 98.2 | 96.6 | 95.9 | 90.3 | 88.7 | 92.0 |
| Single-copy | 46.9 | 51.8 | 53.0 | 61.0 | 59.7 | 49.2 |
| Duplicated | 51.3 | 44.8 | 42.9 | 29.3 | 29.0 | 42.8 |
| Fragmented | 0.4 | 1.0 | 1.6 | 3.2 | 3.9 | 2.9 |
| Missing | 1.4 | 2.4 | 2.5 | 6.5 | 7.4 | 5.1 |

The seventh column provides the statistics for the published draft genome of an invasive lineage, and the sixth column shows the statistics for the published draft genome of the invasive lineage when haplotigs were removed.

deleterious load in the species. This comprehensive study gives deep insights to genome evolution in grass species, and particularly it suggests the specific changes underlying invasiveness, not only in *P. australis*, but possibly also other grasses in general.

## Results and discussion
### Chromosome-level genome assemblies of *P. australis* lineages
Flow cytometry-based estimate of the haploid genome size of the reference *P. australis* (Chinese lineage, denoted as CN) selected for genome assembly was 0.92 Gb (Supplementary Fig. S1), with cytogenetic analysis on cell metaphase revealing 2n = 50 chromosomes. PacBio HiFi sequencing yielded a 61-fold coverage of the predicted genome size with an average read length of 16 kb. Contig-level assembly from Hifiasm was 920 Mb (Table 1), with an N50 of 34.22 Mb. Scaffolding with chromosome conformation capture (Hi-C) data allowed us to map 94.5% of the assembly into 25 pseudomolecules (Fig. 1b; Supplementary Fig. S2; Table 1), and subsequent gapfilling resulted in gap-free pseudochromosome assemblies. Telomeres were detected at both ends of all pseudomolecules, suggestive of complete telomere-to-telomere assembly for all chromosomes (Supplementary Fig. S3). Genome completeness assessed by quantifying the presence of universally conserved single copy genes (BUSCO v5 with Poales odb10 database) suggested a high-quality assembly with 98.9% completeness (Table 1). An intrachromosomal inversion on Chr 22 was detected by phased haplotypes and Hi-C contact maps (Supplementary Fig. S4).

Annotation using a hybrid pipeline[31] combining ab initio predictions, conserved proteins in model species, as well as RNA-seq data yielded 42,498 gene model predictions, with 98.2% BUSCO completeness (Table 1). Altogether 42,369 genes (99.7%) were anchored to the 25 chromosomes (Fig. 1b). Repeat modeling annotated 63.95% of the genome as repeats, with *Gypsy* (10.73%) and *Copia* (7.18%) being the most abundant types (Fig. 1c, Source data https://doi.org/10.6084/m9.figshare.26103652, Supplementary Data 1). We additionally identified 851 tRNA on the 25 chromosomes, with

the highest number detected on Chr 24 (74 genes). Additionally, we predicted 3749 rDNA genes, with most of them located on Chr 24 (776 genes), Chr 17 (579 genes), Chr 20 (403 genes) and Chr 16 (317 genes).

To assess the genomic diversity in *P. australis* we conducted further sequencing on three representative genetically distinct lineages using Illumina short reads : the so-called US land type occurring in Gulf Coast (USland; Y7)[23], native US lineage limited to the north of United States (USnat; Y17)[20] and Mediterranean lineage (Med; Y21)[32]. The draft assemblies exhibited sizes comparable to the chromosome-level reference; 922 Mb for the USnat lineage (Y17), 815 Mb for the Med lineage (Y21), and 1.2 Gb for the USland lineage (Y7). The USland lineage has been hypothesized to be a hybrid between *P. australis* and *P. mauritius*, which would explain its larger genome size[23]. Quality assessment using universally conserved single-copy genes (BUSCO v5.2.1) yielded values similar to the high-quality reference (92.5–98.1% completeness; Table 1), with equal proportions of duplicated genes across all assemblies (34.5–51.7%). These figures were again similar to the number of duplicated genes in the reference and their original contig-level draft assemblies (Supplementary Data 2). Altogether, this suggests that the draft assemblies were of similar quality and accurately captured the gene coding space. Additionally, we included the previously published draft genome of the invasive lineage from North America (EUinvasive)[26] in our analyses. Using the CN genome as a reference, we proceeded to scaffold these four draft assemblies to chromosome-level pseudomolecules. Based on homology, we successfully mapped 94.4–97.1% of the contigs to pseudochromosomes for the draft assemblies of the three main lineages, and 79% for the USland lineage (Supplementary Data 3, Supplementary Data 4). The scaffolded assemblies displayed significantly higher levels of repeat content compared to the contig-level assemblies. As LTR retrotransposons are good indicators of assembly continuity and completeness[33], we assessed the LTR Assembly Index (LAI). The reference genome yielded a relatively high value of 12.71(Fig. 1d, Source data https://doi.org/10.6084/m9.figshare.26104750) while the LAI for the chromosome-

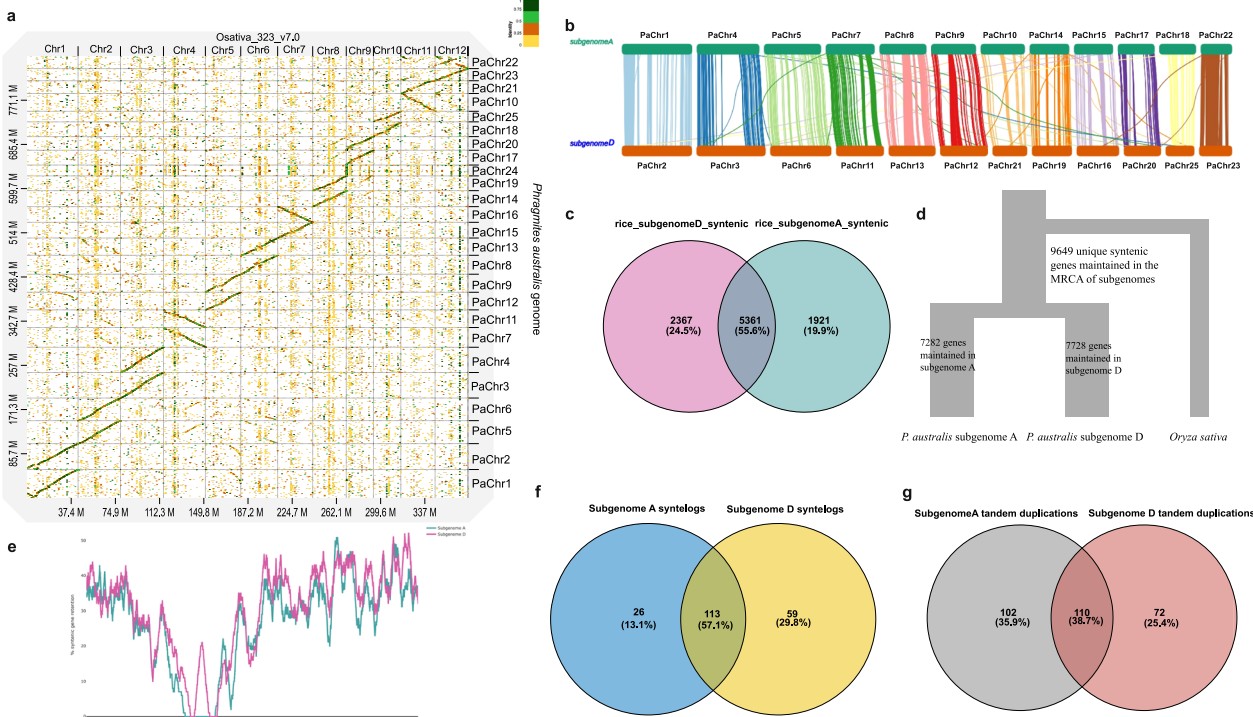

**Fig. 2 | Subgenome synteny, structural rearrangements, gene retention, fractionation bias and tandem duplications in *Phragmites australis*. a** Dotplot of the syntenic regions between *P. australis* and rice (*O. sativa*). The plot shows that two homeologous chromosomes of *P. australis* align to one rice chromosome, representing the two subgenomes of *P. australis*. **b** Large-scale synteny and structural rearrangements between subgenomes A and D of *P. australis* reference genome, with a minimum alignment length of 10,000 bp. **c** Venn diagram showing the number of shared syntenic genes with rice in subgenomes D (left) and A (right). **d** A comparison of the numbers of syntenic gene pairs (syntelogs) with rice illustrates

differential loss rates between the two subgenomes. **e** Fractionation bias of subgenome D over subgenome A in *P. australis* chromosomes homeologous to rice Chromosome 5. The line chart shows the number of retained syntenic genes computed using a sliding genomic window of 100 genes for homoeologous chromosome pairs in *P. australis* and *O. sativa japonica* genome assemblies. **f** A Venn diagram illustrating the number of significantly enriched GO terms and their overlap among subgenome A and D syntelogs resulting from whole genome duplications. **g** A Venn diagram illustrating the number of significantly enriched GO terms and their overlap among subgenome A and D tandemly duplicated genes.

level USnat, Med, USland and EU invasive lineage assemblies was 7.07, 6.97, 6.72, and 9.47, respectively. Since the EU invasive lineage was initially assembled using PacBio long reads, the scaffolding resulted in a higher-quality assembly compared to the contig-level draft assemblies.

## Subgenome phasing reveals subgenome dominance patterns in *P. australis*

Syntenic alignment of the HiFi reference assembly against *O. sativa* revealed a structure of duplicated syntenic blocks, implying the presence of two subgenomes within *P. australis*, characterized by remarkably conserved synteny (Fig. 2a). Subphaser[34] partitioned the genome into subgenomes denoted as A and D, each consisting of 12 pseudochromosomes (Supplementary Fig. S5). Although synteny between the two subgenomes was highly conserved, some large chromosome inversions and minor rearrangements were still identified (Fig. 2b; Supplementary Fig. S6). Out of a total of 42,396 genes on pseudomolecules, 21,152 were assigned to subgenome A, and 20,931 to subgenome D, respectively. The histogram of synonymous mutations (Ks values) between syntenic gene pairs (syntelogs) from self-self alignments for each of the subgenomes indicated three whole genome duplication events in their ancestry, with highly conserved, large syntenic blocks from the most recent event (Supplementary Fig. S7). Compared against the Ks spectrum from syntenic alignments against rice, the modes of the Ks peaks suggested that the events predate the divergence with rice, indicating them to be the pan-Poales *rho*, *sigma* and *tau* WGD events[35]. Assuming that *rho* event occurred at ~70Mya[36] and using this event to estimate a mutation rate, the Ks peak resulting from the syntenic alignment of the two subgenomes suggests their divergence at ~30.9 Mya (recent Ks peaks at 0.2; Supplementary Fig. S8).

Following the allopolyploid event, during the subsequent diploidization process, the genome undergoes fractionation, loss of redundant genes. This loss can occur either in an unbiased random manner, where neither of the genomes dominates the process[37], or it can be biased towards one of the subgenomes, resulting in subgenome dominance[38,39]. In the case of *P. australis*, we observed a significant subgenome bias among the syntelog counts between rice and the two subgenomes (Chi-square test: $p = 4.471e^{-07}$). In total, 2367 rice syntelogs from subgenome D were lost in subgenome A, while, in contrast, 1921 rice syntelogs present in subgenome A were lost from subgenome D. Altogether, we identified 5361 syntelogs retained in both subgenomes (Fig. 2c, d). The same asymmetry was reflected in the BUSCO genes, as a total of 79.6% of single copy genes from the Poales database were identified in subgenome A, while 84.7% were identified in subgenome D, respectively. This suggests that subgenome D dominated over A during genome fractionation subsequent to the allopolyploid event (Supplementary Fig. S9; Table 1). The bias towards subgenome D was more emphasized in nine out of 12 chromosomes in terms of fractionation patterns as well as the numbers of retained syntelogs with rice (Fig. 2e, Supplementary Fig. S10). We next analyzed Gene Ontology (GO) term enrichments for the genes originating from whole genome duplications versus tandem duplications to find out whether these genes are involved in biological processes having adaptive potential. Following the dosage-balance hypothesis[40], the WGD syntelogs were mostly enriched for regulatory activities such as regulation of biological process, various metabolic processes, regulation of gene expression, developmental process, and biosynthesis in both subgenomes (Fig. 2f, Supplementary Fig. S11, Supplementary Data 5). The accumulation of tandem duplications is an evolutionarily more rapid mechanism and, accordingly, these genes were

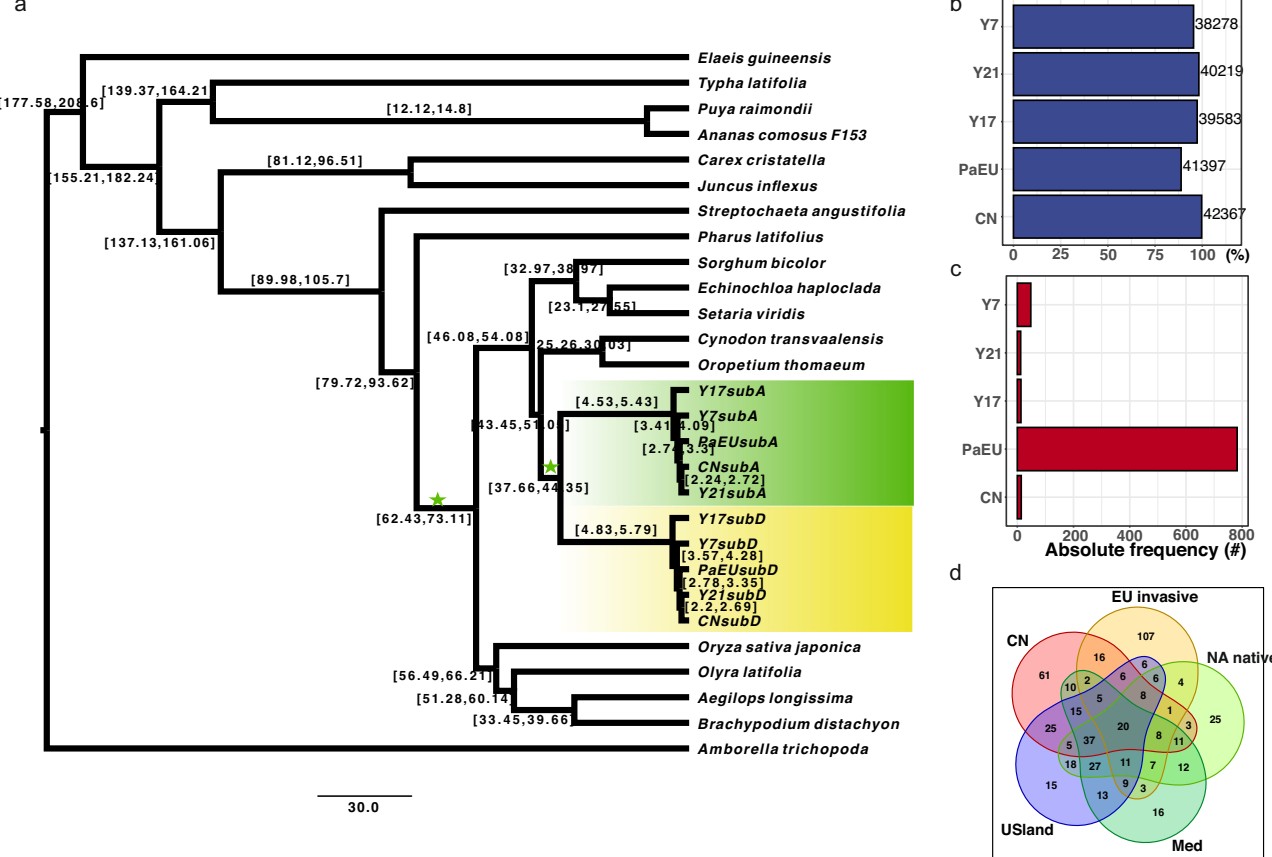

**Fig. 3 | Comparative genomics of *Phragmites australis*.** Analysis of subgenome divergence, phylogenetic relationships, distribution of orthologous genes, and patterns of tandem duplications. **a** A phylogenetic tree estimated from 189 single copy BUSCOs of 23 species, with *Amborella trichopoda* as an outgroup. The subgenomes of each *P. australis* lineage were fully phased and used separately in estimation. The 95% confidence interval (CI) is shown in the square brackets along each of the branches, with Million Years ago (Mya) as unit. **b** The proportions (x-axis) and the numbers of genes (next to bar plots) clustered into orthogroups for the five *P. australis* lineages. **c** The number of genes in lineage-specific orthogroups for the five *P. australis* assemblies. **d** A Venn diagram illustrating the number of shared and lineage-specific Biological Process gene ontology (GO) terms among tandem duplications in each of the lineages. Unlike other lineages, tandem duplications in the EU lineage were significantly enriched for 14 GO categories, primarily associated with DNA structure maintenance and replication.

mostly enriched for response to stimulus, response to stress and other organisms, as well as defense responses (Fig. 2g, Supplementary Fig. S11; Supplementary Data 6).

### Common reed lineages originate from an ancestral hybridization
Genome assembly and population analysis of Arundiaceae species has historically been challenging due to high ploidy level variance. To assess phylogenetic placement and divergence times between the subgenomes, we identified 189 orthologous genes present in 19 species as a single copy but duplicated in all *P. australis* assemblies, with one copy per subgenome (Supplementary Data 7). Using *Amborella trichopoda* as an outgroup, the phylogeny estimated from this set grouped the *P. australis* lineages as a monophyletic clade, with the North American lineage sister to all other lineages (Fig. 3a). According to our calibrated trees, with two calibration points including the common ancestor of BOP clade (69.6–81.9 Million years ago, Mya) as well as the divergence time between monocots and dicots (179.9–205 Mya), the two subgenomes of *P. australis* appeared to diverge at an early age (37.7–44.4 Mya). This estimate appears to be much earlier than ~17 Mya reported by Huang et al. [41] who used several additional secondary calibration points and relatively similar to the ~30.9 Mya obtained with Ks peak comparisons above. The differences may stem from the conservative selection of calibration points in our data and a fixed molecular clock, varied species diversification rate among branches that could affect phylogenetic estimates, or purifying selection acting on the highly conserved BUSCO genes while being more relaxed with the newly duplicated genes [42]. Both subgenomes yielded concordant phylogenies for the *P. australis* lineages, following the divergent order of USnat, USland, EUinvasive, CN and Med lineages (Fig. 3a), with the lineage splits occurring during the last ~5 Mya.

### Gene families involved in DNA replication are expanded in the invasive lineage
To investigate gene family evolution specific to the invasive lineage, we clustered the proteomes of the five representative *Phragmites* assemblies using Orthofinder. The assemblies yielded similar numbers of genes in the orthogroups (Fig. 3b), with the EUinvasive lineage showing the highest number of lineage-specific orthogroups and duplications (Fig. 3c; Source data: https://doi.org/10.6084/m9.figshare.26356390). The genes encoding the proteins in the EUinvasive-specific orthogroups were enriched for multiple biological processes including telomere organization, telomere maintenance, DNA geometric change, DNA replication, chromosome organization, recognition of pollen, DNA metabolic process and DNA biosynthetic process (Supplementary Data 8). These processes are closely related to proliferative activities such as the growth rate, the accumulation of biomass, and the balanced allocation of energy to sexual and asexual reproduction of common reed.

The gene family expansions as such provide no direct evolutionary interpretation of the adaptive potential in the invasive lineage, and furthermore, the results could be susceptible to the particular clustering approach used. Dosage increases in the form of tandem duplications have been proposed as the general rapid evolutionary mode of adaptation [43,44], and

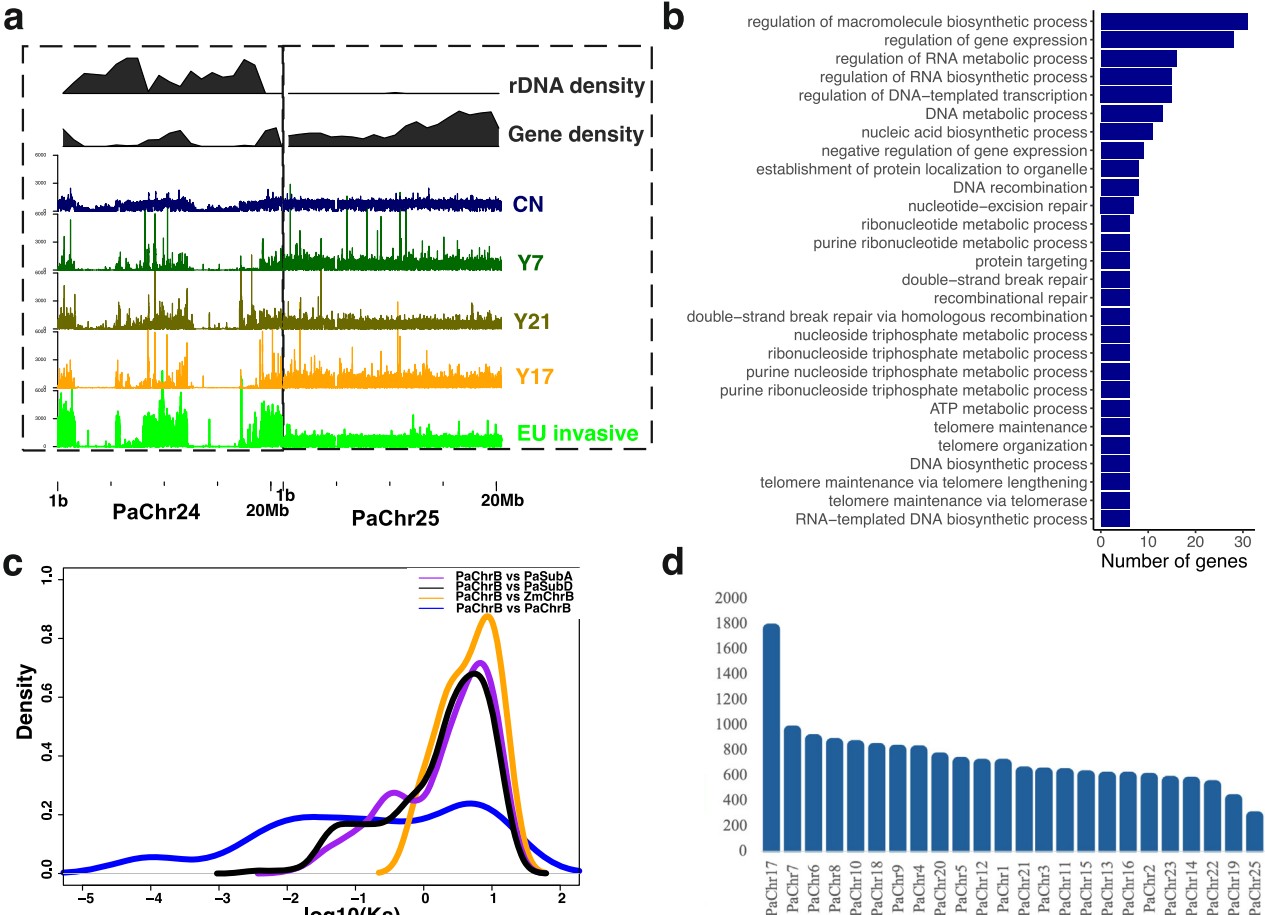

**Fig. 4 | Genomic insights into the B chromosome of *Phragmites australis*.**
**a** Sequencing coverage over B chromosomes. The tracks (from top to the bottom) illustrate the rDNA density, gene density, followed by average read coverage on CN, USIand (Y7), Med (Y21), USnat (Y17), and EU invasive individuals, when mapped on the B chromosome reference (Chr 24; left) or a non-B chromosome Chr 25 (right window). Dashed line indicates the borders of the chromosomes, while the *y*-axis illustrates read coverage. The invasive individual exhibits notably higher read coverage on the B chromosome compared to Chr 25. **b** The Gene Ontology (GO) terms significantly enriched among the genes on the B chromosome. The enrichment

analysis was carried out for GO Biological Processes categories, and the significances were adjusted for multiple comparisons using Bonferroni correction. **c** Density plot of the synonymous substitution rate (Ks) distributions of the syntelogs between the B Chromosome and subgenomes A (purple) and D (black) of the main genome. Additionally, the panel shows Ks plots of *P. australis* B chromosome against the maize B chromosome (gold) and B chromosome against itself (blue). **d** A bar plot of the numbers of homologous fragments (300 bp size) between the B chromosome and *P. australis* main genome.

therefore we carried out a complementary analysis for the tandemly duplicated genes to test the hypothesis. Overall, the highest amount of tandem duplicates were identified in the EUinvasive lineage (13,114 genes), followed by the reference CN assembly (7572 genes), while the short-read assemblies demonstrated somewhat lower numbers (5657, 5950, 6162 duplicates), likely due to their lower contiguity. However, these sets were clearly large enough and of comparable sizes to allow comparison of process-specific expansions between the lineages. A majority of the enriched gene ontology categories demonstrated overlap across the lineages, but we additionally detected 14 GO processes comprising 255 genes that were specific to the invasive lineage and, intriguingly, most were related to DNA replication and telomere maintenance (Fig. 3d, Supplementary Data 9). Tandem duplications in plant species have been generally found to play a role in environmental responses[43], whereas in invasive common reed there seems to be a greater emphasis on maintaining DNA integrity.

To test whether the gene family expansions detected in the clustering approach were due to hybridization or adaptation in the form of tandem duplications we tested the overlap between the two approaches. The overlap was significant ($p < 3.7e^{-62}$; Fisher Exact test) with 42.5% of the expanded genes being present in tandem duplications. Altogether, the results suggest that lineage-specific gene family expansions are associated with the well-

documented increased proliferation in the invasive lineage[45] and that they at least partly result from rapid adaptation in the form of increased dosage through tandem duplications. Considering the young evolutionary age of the invasive lineage this expansion has been relatively rapid.

### *Phragmites australis* has a large B chromosome
Interestingly, karyotyping of the reference individual identified 25 diploid chromosomes instead of the expected 24, leading us to suspect that the reference individual possessed a B chromosome. While the assembly size of Chr 24 was relatively large in the reference CN lineage (21 M), its size was much smaller in the other assemblies (Supplementary Data 4). We speculated that this discrepancy might be due to high repeat content that gets assembled accurately with the PacBio HiFi, while the other accessions would either contain collapsed assemblies (due to platform limitations) or no assembly at all due to the absence of B chromosome in those individuals. When the short reads were mapped against the scaffolded chromosome-level assembly from the same individual, the read coverage on Chr 24 was consistently higher than on other chromosomes, suggesting collapsed contigs in these draft assemblies. Mapping against the reference assembly largely resolved these issues, as the coverages became more similar to the overall genome coverages (Fig. 4a). Elevated coverage was still observed in

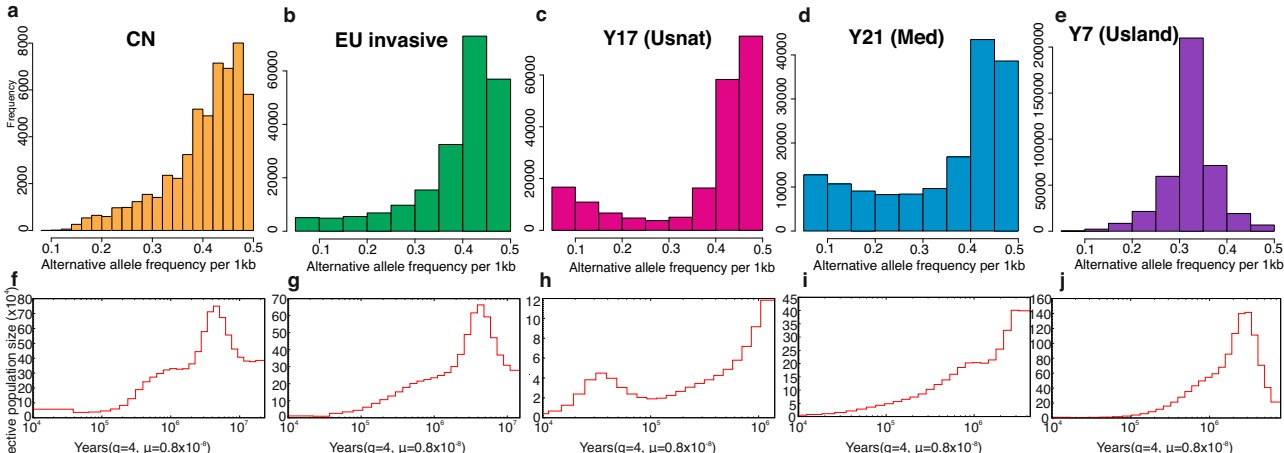

**Fig. 5 | Ploidy levels and historical effective population sizes of *Phragmites australis* accessions.** The figures in the upper panel show the inferred ploidy levels for each of the accessions. The estimates were obtained by aligning short-read sequencing data against the reference genome. Geographic lineages CN (**a**), EU invasive (**b**), USnat lineage (Y17; **c**) and Med lineage (Y21; **d**) were inferred to be allotetraploid with a minor allele frequency (MAF) distribution peak at 0.4–0.5. The MAF for the USland type of Gulf Coast (Y7; **e**) was 0.3–0.35, indicating that this individual is allohexaploid. The lower panel shows historical effective population sizes for these accessions (**f**: CN, **g**: EU invasive, **h**: Y17, **i**: Y21, **j**: Y7). The demography was estimated from short-read sequences mapped to their respective assemblies and using Pairwise sequential Markovian coalescent (PSMC) model.

the North American invasive lineage, suggesting aneuploidy for this chromosome, while USnat and USland demonstrated lower coverages.

A closer analysis of Chr 24 revealed high repeat contents (89.8%), along with a redundant accumulation of non-coding RNA genes such as rDNA (776 genes) and tRNA (74 genes), and a lack of large syntenic blocks with any of the other chromosomes in *P. australis*, hence we may conclude it to be a B chromosome[46]. In *Arabidopsis* the length of rDNA clusters can reach up to millions of base pairs[37,47], and copy number variation in multiple species of hybrid origin such as Arabidopsis[37], strawberry[48], and synthetic wheat[49] for these loci have been identified. Therefore, the apparent copy number variation of B chromosome among *P. australis* lineages may also be confounded by variation in rDNA amounts. However, an in-depth analysis of the read alignments showed mappings on the B chromosomes mainly on gene rich regions rather than on rDNA regions, suggesting that the variations in coverage were not due to rDNA (Fig. 4a). To estimate the B chromosome copy number in the different accessions we calculated the ratio of averaged read coverage on B chromosome versus on all other chromosomes, identifying a much higher value for the invasive lineage (2.29), and medium value for CN (0.93) and Med lineages (1.01), but only 0.36 for USnat (Y17) and 0.29 for USland (Y7). This suggests that the sequenced invasive accession likely contains a higher number of B chromosomes (2x) than the reference, while the low ratio in Y17 and Y7 suggests that B chromosome may not exist in these individuals.

B chromosomes have been reported to evolve by accumulating genes from A chromosomes (with A denoting the standard chromosome set), as gene fragments or DNA homologous to the A chromosomes have often been detected[50]. In total, 286 protein coding genes were predicted on the B chromosome. These genes were enriched for biological process GO terms such as fusion of sperm to egg plasma membrane involved in double fertilization forming a zygote and endosperm, chromosomal segregation during cell cycles, telomere maintenance, and associated biochemical activities such as ATP synthesis and metabolism, protein localization, DNA synthesis and repair, attachment of spindle microtubules to kinetochore in mitotic and meiotic cell cycles (Fig. 4b; Source data: https://doi.org/10.6084/m9.figshare.26356621), potentially contributing to increased proliferation.

To study whether the genes in B chromosome were expressed in the different accessions, we reanalyzed the transcriptomic data from Oh et al.[26]. A mapping of RNAseq reads onto the reference genome suggested that genes on the B chromosomes were expressed in all tissues of all individuals. However, since the number of B chromosome copies in the individuals used for RNAseq was not known, the presence could also be attributed to recent duplications of homologous genes on other chromosomes. We found the highest number of unique transcripts expressed in the shoot tissue of the invasive lineage (Supplementary Data 10), supporting the possible higher copy number at least in the invasive lineage.

To determine the duplication modes of B chromosome genes, we identified regions that were syntenic with subgenomes A and D and determined their synonymous substitution rates (Ks). When filtering for genes with high Ks values (Ks>5) to exclude artefacts from synteny alignment or errors from gene prediction, we detected altogether 254 and 284 genes in subgenomes A and D, respectively. A histogram of Ks values suggested the B chromosome duplicates to be of ancient origin, with the peak of Ks values comparable, or possibly even older in time, than the *tau* whole genome duplication event common to monocots. The Ks values displayed a similar peak when aligning this B chromosome against the B chromosome in *Zea mays* (Fig. 4c) with default parameters and then reducing the syntenic block length to one gene. According to *P. australis* self-self alignments the B chromosome had undergone two subsequent, possibly segmental, duplication events (Fig. 4c). Together, this suggests that even though rapidly evolving, the B chromosomes across different species may have a common origin, and the core set of genes has been shaped by repeated transfer of genetic material from the main chromosomes.

To further inspect the origins of the B chromosome outside of the gene coding regions, we divided the chromosome into 300 bp sequence fragments and mapped them against the main chromosomes. Since the unanchored contigs may be part of B chromosome or other chromosomes, we only targeted chromosomal level scaffolds to ensure the accuracy of tracing the origins. Out of the 70,085 fragments, 17,013 mapped primarily, suggesting that at least 24.27% of the B chromosome has been sourced from the main chromosomes. Of these, 1758 sequences mapped to pseudochromosome Chr 17, while 953 sequences mapped to Chr 7(Fig. 4d; Source data: https://doi.org/10.6084/m9.figshare.26356660).

## Complicated life history involved in common reed evolution

To assess the impact of polyploidization events on intraspecific species divergence, we estimated the ploidy level of each individual using the ratio of reads supporting the minor allele versus all reads aligned to a locus[51]. We followed the logic that in a diploid (or, in this case, an allotetraploid with clearly separating subgenomes) genome, the number of reads supporting the minor allele in a heterozygous locus should be half of the total read

**Table 2 | Nucleotide diversities at neutral and deleterious sites in five different *P. australis* accesions**

| | Nonsynonymous π | Synonymous π | $\pi_{N/S}$ |
|---|---|---|---|
| Y17 | 0.001417 | 0.001142 | 1.24 |
| Y21 | 0.002784 | 0.003121 | 0.89 |
| Y7 | 0.008596 | 0.009470 | 0.91 |
| PaEU invasive | 0.004491 | 0.007634 | 0.59 |
| CN | 0.004636 | 0.013686 | 0.34 |

The diversities were estimated separately for each assembly using gene models transferred from the reference assembly.

count. Likewise, in a triploid genome, the proportion of reads mapping to minor allele should be one third. The scaffolded reference assemblies for CN, North American invasive, Med and USnat lineages all displayed minor allele frequency peaks at 0.4–0.5 (Fig. 5a–e; Source data: https://doi.org/10.6084/m9.figshare.26356684), suggesting that reads are uniquely mapping to either of the two subgenomes. This observation aligns with the notion of common reed being an allotetraploid, exhibiting two highly differentiated subgenomes. Additionally, a peak at 0.33 was observed for USland, suggesting a hexaploid genome organization. The higher ploidy level is also supported by the 1.5x larger assembly size compared to others and a large proportion of contigs that were not mapped to the pseudomolecules (Table 1).

Next, we modeled demography using the pairwise sequentially Markovian coalescent (PSMC) model, using four years as generation time. PSMC analysis showed consistent decreases in CN, EU invasive and Med lineages (Y21), all starting from around 4 Mya. In USland, the decrease started from around 3 Mya, and in USnat from 1 - 2 Mya, with the latter demonstrating a slight recovery at 30–40 Kya followed by another decline (Fig. 5f–j).

The decreasing population sizes may be related to biogeographic events in Pliocene, such as glaciation cycles. Another possibility is a transition into asexual reproductive mode. Population genetic simulations predict that compared to outcrossing and selfing lineages, a clonal lineage experiences relaxed selection and thus accumulates more deleterious mutations[52]. Contemporary common reed uses both sexual and asexual reproductive modes. The seed set rate of *P. australis* is very low (mean 9.7–12%) due to its partial self-incompatibility[53,54] while, in contrast, clonal propagation has been shown to be highly efficient[55–57]. Due to the lack of recombination, genotypic diversity would be expected to be low in clonal propagation since genotypes that have accumulated a high number of deleterious alleles, and therefore have reduced fitness, will be purged from the population (the so-called Muller's ratchet). Additionally, clonal populations may undergo frequent bottlenecks, as they can colonize habitats with just a few individuals. Altogether, these effects result in effective population decline in the long run[58]. These predictions have been empirically verified in both animals and plants, such as *Primula* species[59], stick insects[60], duckweed[61], and wasps[62]. Relaxed selection would result in an increased ratio of nonsynonymous to synonymous mutations in asexual and selfing proliferative strategies, but a low ratio in outcrossing species because purifying selection would limit the amount of deleterious alleles in outcrossers[58].

In support of this hypothesis, nucleotide diversities of deleterious sites ($\pi_n$) were low for USnat (Y17; $\pi_n = 0.001417$) and Med Y21 ($\pi_n = 0.002784$), while being higher for USland Y7 ($\pi_n = 0.008596$), EU invasive ($\pi_n = 0.00449$) and CN ($\pi_n = 0.00464$) (Table 2). To assess the proportion of deleterious load we compared their ratios to synonymous diversity and observed high values for USnat ($\pi_n/\pi_s = 1.24$), followed by 0.89 for Med and 0.91 for USland. For EU invasive lineage the proportion was considerably lower, 0.59, and 0.34 for CN. Altogether, the ratios are high compared to other plants[63], and can be interpreted to reflect predominantly clonal propagation, since empirical studies and theory have shown that clonal propagation accumulates more deleterious mutations due to relaxed purifying selection, resulting in higher $\pi_n/\pi_s$ ratio than among outcrossers[64,65]. The proportion was highest in USnat population, which has long been isolated

from other lineages. The population statistics for the USnat thus possibly result from a combination of clonal propagation (high $\pi_n$, high population level $\pi_s$), inbreeding depression (low individual level $\pi_s$), and little gene flow from other *P. australis* populations (early divergence in species tree). In addition, it has been found that in North America, the long-distance dispersal of USnat is facilitated by seedlings, but local growth is mainly proceeding through vegetative propagation[66]. Interestingly, the ratios in invasive lineage show a decrease in the proportion of deleterious alleles, suggesting that selection is more active in this population. Therefore, asexual propagation through cloning may accumulate the mutational load, reduce the fitness for the lineages and result in decreased population sizes[67].

## Conclusion

We present a chromosome-level genome assembly for an invasive grass species *P. australis*, and chromosome-level scaffoldings of four draft assemblies representing distinct lineages. This genome assembly marks a major advancement in our understanding of the Arundinoideae species renowned for their complex genomic history[68]. These species may have undergone multiple whole genome duplication events and hybridizations, rendering the tracing of their origins challenging[68]. Utilizing the unique repeat patterns of each parental genome, we phased 24 chromosomes of common reed to two subgenomes. Our analysis revealed biased subgenome fractionation following the allopolyploid event, with subgenome D playing a more dominant role. Phylogeny estimated from phased subgenomes of the five representative lineages and the divergence time estimate from Ks values indicate that the allotetraploid *P. australis* has formed as a hybrid of two ancestral closely related species at around 30.9 million years ago. Hybridizations between *Phragmites* species may still occur and can lead to higher ploidy levels within the species, as evidenced by the hexaploid USland lineage[69]. Therefore, hybridization and polyploidization are essential drivers of genome evolution in grass species, but they are not direct causes of invasiveness, as both North American native and invasive lineages are allotetraploids. Decreasing effective population size and high deleterious load suggests that *P. australis* is going through a transition of reproduction mode from sexual to asexual propagation which is one of the major features in invasive plants.

North American invasive accessions consistently outperform North American native and European native lineages in terms of shoot biomass[45]. While hybridization with a yet unknown *Phragmites* species cannot be ruled out, this difference could also be linked to higher B chromosome copy numbers in the invasive lineage, as observed for the single representative in this study. B chromosomes potentially promote the adaptability of some plant species; for example, individuals with B chromosomes have demonstrated a higher survival rate under stress in *Allium schoenoprasum*[70] and *Lolium perenne*[71]. B chromosomes also affect phenotypic traits and gene expression of A chromosome genes, with a more emphasized effect in the presence of more copies[72–75]. In maize, thousands of genes on A chromosomes were upregulated in the presence of B chromosomes, while the B chromosome gene expression was positively correlated with its copy number in a dosage-specific manner[76]. However, exceptionally high copy number of B chromosomes may also reduce the fertility and viability of offspring, thus restricting sexual reproduction[77].

Our study also demonstrated lineage-specific gene family expansions, particularly tandem duplications, as potential contributors to invasiveness; similar effects have been observed in a malignant invasive plant *Heracleum sosnowskyi*[78]. The expanded gene families were mostly enriched for DNA replication and telomere maintenance, suggesting their essential role in maintaining genome integrity as well as DNA synthesis, thereby enabling fast regeneration and proliferation.

In conclusion, we suggest that common reed may have gradually acquired traits facilitating invasiveness, although the observed invasion has occurred within a short time frame. Future controlled experiments are needed to validate the function of these genetic elements in common reed, and to study their associations with physiological traits at the population level. Ultimately, this will contribute to a greater understanding of the decisive factors in grass invasiveness.

## Methods

### DNA extraction, library construction and whole genome sequencing of the reference individual

For the reference genome sequencing, young leaves were sampled from an individual originated from Yuncheng, Shanxi province. DNA extraction was done using Qiagen DNA extraction kit. One paired-end library was constructed for short-read sequencing on BGI T7 sequencing platform in Hongshu Shanghai limited company, which generated around 230 G (255x) of high-quality data. High Weight Molecular DNA was extracted by Biozeron Biotech Ltd. High fidelity long reads were obtained from Pacific Biosciences Sequel IIe platform, with about 61-fold coverage. Long DNA of the same individual were used for Hi-C library preparation. After crosslink, cut with DpnII enzyme, biotin labeling, ligation, shearing and biotin capture, the DNA fragments were sequenced using Illumina Novaseq platform, with an average depth of 110-fold across the genome. Transcriptomic RNA was extracted from the same individual to help with the prediction of gene models.

### Genome size estimation and cytogenetic confirmation

To estimate the genome size of the reference individual, we used flow cytometry method. Briefly, plant tissue was initially digested with pre-cooled MG solution (45 mM $MgCl_2 \cdot 6H_2O$, 20 mM MOPS, 30 mM Sodium citrate, 1% (W/V) PVP40, 0.2% (v/v) Tritonx-100,10 mM $Na_2EDTA$, 20 μL/mL β-mercaptoethanol, pH 7.5). The tissue was then broken apart thoroughly using sharp blade and placed still on ice for 10 min, followed by filtration using 40 μm diameter mesh to obtain the cell suspension. Add 50 μg/mL propidium iodide and RNAase, then place the cell suspension on ice in dark for staining. We selected *Zea mays* of which the haploid genome content is 2.3 Gbp as an internal reference and used BD FACS calibur flow cytometer to detect the fluoresce intensity of the stained cell nucleus suspension. To count the number of chromosomes of the reference individual, we collected roots and treated it with HCl: ethanol (1:1) under 65 °C for 10 min, wash the treated tissue in deionized water for 5 min, stained the tissue with Carbol fuchsin, squeeze it on a glass slide, and observe the metaphase under microscope.

### Assembly of organelle genomes and nuclear reference genome

The Pacbio HiFi reads were first used to perform a de novo assembly of chloroplast genome for *P. australis* using PMAT[79]. The mitochondrial genome was assembled with Flye[80] and extracted using GetOrganelle[81] based on the embplant_mt database. The resulting circular organellar genomes were visualized and corrected using Bandage software[82]. After removing the Pacbio HiFi reads that were primarily mapping to the organelle genome, the remaining Pacbio HiFi reads were assembled using Hifiasm[83], resulting in a primary phased haplotig assembly with genome size 920 Mb in 1350 contigs. The draft genome was then scaffolded to pseudomolecules using HiC contact reads and ALLHiC pipeline[84]. To verify the accuracy of scaffolding, the HiC reads were also mapped to the draft genome using juicer[85], and the ALLHiC result was then converted to the format that could be visualized in juicebox[86] for manual correction to improve the contig orders and chromosome boundaries. The final assembly was obtained from the postreview procedure in 3D-DNA pipeline[87], and finalized with gap closing using TGS-GapCloser[88].

### Genome assemblies of USland, USnat and Med lineages

Based on previous biogeographic lineages identified using chloroplast markers, we selected three individuals (Y7, Y17, Y21) representing USland, USnat and Med lineages, respectively, to be sequenced for whole genome assemblies. Following DNA extraction with CTAB protocol, the genomes were sequenced to 90-fold high coverage (assuming 1 Gb genome size) by NovogeneAIT at Singapore using illumina Novaseq platform. The genome size was first estimated using KmerGenie[89]. Then, after quality assessment of the reads using FastQC (https://www.bioinformatics.babraham.ac.uk/projects/fastqc/), three draft genomes were assembled using MaSuRCA assembler[90] with default settings. Assemblies based on short reads may

capture gene coding space to a lower accuracy, for example due to collapsed assemblies or separate haplotig assemblies. To avoid these biases, we purged the draft assemblies for haplotigs, thus obtaining similar coverages across the contigs. The assembly was passed through purge_haplotigs v1.0.4[91] to remove the redundant haplotigs resulting from the highly heterozygous regions. The completeness of the genome assemblies was assessed using BUSCO v5[92] using odb_database version 10. The quality of the assembly was estimated using Quast[93]. Next, we scaffolded to pseudomolecules with RagTag[94], which aligns the contigs against the chromosome-level reference assembly. LTR Assembly Index (LAI) was calculated using LTR_retriever[95].

### Reference genome annotation

Repeat libraries were built with repeatmodeler[96], and the repeats were masked using RepeatMasker (Smit, AFA, Hubley, R & Green, P. Repeat-Masker Open-4.0. 2013–2015 http://www.repeatmasker.org). Telomeric repeats were detected using telomeric-identifier (https://github.com/tolkit/telomeric-identifier). The tRNA was predicted using tRNAscan-SE[97]. We identified rRNA and non-coding RNA sequences using Infernal[98].

Gene prediction was performed by combining ab initio prediction, genome homology to close related species, and gene structure prediction from RNAseq evidence. Ab initio gene prediction was performed using GeneMark-ES[99], BRAKER[100], PASApipeline[101] and AUGUSTUS[102]. Annotation information of *Sorghum bicolor*, *Setaria viridis* and *O. sativa* were obtained from Phytozome[103] and used for homology-based annotation using GeMoMa[104]. To improve the gene model predictions, we incorporated 20 RNAseq libraries for genome annotation, including 13 RNAseq samples downloaded from NCBI (Supplementary Data 10). Some of these data were originally used to annotate the genome assembly of the EU invasive lineage[26]. Additionally, we sequenced seven RNAseq libraries representing three different genetic lineages (CN, Med, and South Africa)[16,69]. These reads were aligned against the reference genome with STAR aligner v2.7.11b[105], and used as evidence in BRAKER[106] prediction. In addition, RNAseq data was also assembled into genome-guided assembly using Trinity[107] and stringtie[108], which was further processed using PASApipeline[109], and TransDecoder (Haas, BJ. https://github.com/TransDecoder/TransDecoder) for generating a high-quality dataset for downstream ab initio gene predictions. All the evidence created with different methods were combined using EVidenceModeler v2.1.0[110]. Functional annotation was performed by aligning the proteins against Uniprot and NCBI nr databases using DIAMOND v2.1.8[111], with parameters -evalue 1e-5 -max_target_seqs 5. The completeness of predicted gene models was assessed using Benchmarking Universal Single Copy Orthologs v5.1.2[112] by searching against poales_odb10 protein database with 4650 conserved single-copy genes. Significance of overlap between duplicated BUSCOs in different genome assemblies were tested with Fisher's exact test using R package "GeneOverlap"[113]. Repeat counts and their divergence to the consensus repeat sequences were plotted using repeatR package (https://github.com/dwinter/repeatR).

### Five genome assemblies and subgenome phasing

With the Chinese assembly (CN) as a reference, we scaffolded the draft assemblies of Y17, Y21, Y7 and the published EU lineage to chromosome level using RagTag[94]. For the reference genome, we detected syntenic blocks among each pair of homoeologous chromosomes by aligning them against the rice genome assembly (*O. sativa japonica* v7.0, downloaded from Phytozome) using minimap2 and visualizing the dotplot created using D-GENIES[114]. Based on this, we initially phased the largest 25 chromosomes (except Chr 24) to two subgenomes (A, D). The chromosome assignment to each subgenome was then refined using SubPhaser[34], which calculates subgenome specific repetitive kmer, with the parameter set -k 13 -q 100 -f 2. The visualization of collinearity was performed using NGenomeSyn[115] and Syri[116]. Accordingly, subgenomes of all the four extra geographic lineages were phased to chromosome level, for each subgenome the completeness was assessed using BUSCO v5.1.2. In addition, the five *P. australis* draft genome assemblies were aligned against *O. sativa* to get the synonymous

substitutions for further investigation of multiple whole genome duplication events. Gene models for the three synteny-based chromosome level assemblies were predicted using GeMoMa pipeline, with the annotation of the reference assembly as a reference. Gene models for the EU invasive lineage was obtained from Oh *et al.* 2022, with haplotigs purged. To evaluate the gene loss along each of the homologous chromosome pairs in the two subgenomes, we performed genome fractionation analysis in CoGe platform using FractBias tool by aligning the *P. australis* reference genome against rice (*O. sativa japonica*). For each rice chromosome, the percentage of gene retention within each window iteration (100 genes) from *P. australis* homologous chromosome pairs was displayed in a graph. We evaluated the bias in syntelog losses between the two subgenomes by selecting the syntelogs with Ks value smaller than 0.7. Significance of subgenome fractionation bias was assessed using R package GeneOverlap[113]. The chromosome rearrangements in subgenomes were detected using syri[116], and plotted using plotsr[117]. Tandem duplications and syntenic genes of each subgenomes were analyzed for GO term enrichment using goatools[118]. To find out lineage specific gene families, we performed all-versus-all blast search for the reference proteome and the predicted proteomes form the four contig-level genome assemblies using OrthoFinder v2.5.5[119], and visualized the results using R package cogeqc[120].

### B chromosome copy detection

To analyse the copy number of B chromosomes in *P. australis* lineages, we calculated the read depth within 1 kb window sizes for each chromosome with repeats masked. Several regions were found to have high depth of coverage (Chr14 and Chr 25), peaking in all the five lineages, which may indicate chloroplastic or mitochondrial DNA, or assembly errors. We also masked these regions to avoid noise. We then calculated the ratio of read depth mapped to B chromosome and the average depth of other chromosomes in all the five individuals, assuming the copy number of B chromosome in an individual should be proportional to the ratio of the read coverages. The read depth across chromosome 24 and Chromosome 25 were visualized using R package karyoploteR[121].

### Grass genome evolution

To investigate the parental species and allopolyploidization of *P. australis*, we used 192 single copy orthologous genes from BUSCO evaluation to estimate a phylogenetic tree for 23 plant species including the subgenomes of five *P. australis* assemblies. Species *Amborella trichopoda var. Santa Cruz* was used as outgroup. The gene trees were inferred by maximum likelihood method using RAxML[122] with 100 bootstraps, and combined into species tree using Astral-III[123]. To help with dating the time of allopolyploidization, we estimated divergence time using mcmctree[124] in PAML[125], with most common ancestor of BOP clade (69.6–81.9 Mya) as well as the divergence time between monocots and dicots (179.9–205 Mya) as calibration point[126,127]. Two runs of mcmctree runs were compared to check whether convergence was reached.

### Ploidy level prediction

Since the number of alternative alleles other than the reference allele in the genome can reflect the copies of homologous chromosome number[128], we predicted the ploidy level information using minor allele frequency (MAF) after mapping the reads to the reference genome. By setting the total read coverage to range from 20 to 200, only alleles with coverage higher than 7 were considered as variants. For example, in diploids or allotetraploid in current study, each genomic region has two copies of alleles, the proportion of reads supporting alternative alleles would be around 0.5 in heterozygous sites, and thus the genomic distribution of the proportion of reads supporting alternative alleles should demonstrate a single peak at 0.5. Similarly, for an allohexaploid individual, the minor allele frequency should peak at around 0.33 etc. Based on this pattern, we predicted the ploidy levels by plotting the density of MAF and compare the predictions using the flow cytometry data.

### Population demographics in history

We aligned the reads of the five whole-genome-sequenced individuals to their own reference genome using bwa[129], and performed demographic analysis using PSMC[130]. Although it usually takes 1 year for *P. australis* to sprout and mature, we set the generation time to 4 years because it is perennial. We set the mutation rate to be the Poaceae silent-site mutation rate, $8e^{-09}$ substitutions per site per year[131].

### Identification of putative deleterious mutations

The nucleotide diversities of neutral and deleterious sites were calculated following the procedure in https://github.com/jsalojar/PiNSiR. Briefly, the short reads of Y17, Y21 and Y7 were aligned to their own draft genome using bwa mem[129] to obtain the variants along the genome, sorted with samtools[132], and genome coverage variants were called using bcftools[133]. Site-wise diversities were estimated using ANGSD[134] and used as input in the R package PiNSiR. The deleterious mutations were determined using SnpEff[135], and assuming the loci with "HIGH | MODERATE" flags to be deleterious.

### Statistics and reproducibility

The gene overlap tests were carried out with either Fisher's exact test or Chi-square tests. The GO enrichment was tested using Fisher's exact test followed by *p*-value correction for multiple testing using Bonferroni, Sidak, Holm, and false discovery rate adjustment. The analyses in this study were carried out with genome assemblies from five individuals.

### Reporting summary

Further information on research design is available in the Nature Portfolio Reporting Summary linked to this article.

### Data availability

The raw sequencing data are available in NCBI SRA database under the BioProject ID PRJNA849004. This reference genome assembly has been deposited at DDBJ/ENA/GenBank under the accession JBCHVN000000000. The version described in this paper is version JBCHVN010000000. The assemblies of all the accessions and their annotations are available at CoGe (https://genomevolution.org/coge/): the chromosome-level reference CN individual (id 67975), scaffolded North American invasive lineage (id67997), Med lineage (Y21, id68003), USland lineage (Y7, id68004), and USnat lineage (Y17, id68002). In addition, the assemblies, the annotations, chloroplast and mitochondrial genome assemblies as well as the numerical source data underlying the graphs and charts can all be downloaded from the collection in Figshare (https://doi.org/10.6084/m9.figshare.c.7368682.v1).

### Code availability

The scripts used for the construction of genome assembly and genome annotation, as well as data analysis in this article could be found from github https://github.com/smallfishcui/Phrag_Genome, and zenodo (https://doi.org/10.5281/zenodo.12567126).

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

## Acknowledgements

This study was supported by Young Scientists Fund of Shandong Provincial Natural Science Foundation ZR2021QC119 (to C.W.), International

Postdoctoral Exchange Fellowship Program of China Postdoctoral Science Foundation (to C.W.), National Natural Science Foundation of China U22A20558 (to WH.G.), National Natural Science Foundation of China 32100304 (to LL.Liu), National Natural Science Foundation of China 31800299 (to T.W.), Academy of Finland 318288 (to J.S.), Academy of Finland 319947 (to J.S.) and Nanyang Technological University startup grant (to J.S.). We would like to thank Pasi Rastas for providing the script to calculate alternative allele count, and Sitaram Rajaraman for sharing the work pipeline of genome annotation. We appreciate the assistance from James Ord in improving the English language throughout this article. Finally, we want to acknowledge CSC–IT Center for Science, Finland, and NTU HPCC, Singapore, for computational resources.

## Author contributions

C.W., W.G. and J.S. conceived the study; H.B., L.L., F.E., M.Y., Y.W. and T.W. maintained and collected the samples as well as provided sampling information; C.W., L.L. and M.Y. performed DNA extraction; C.W. and J.S. lead the data analysis; L.L. performed the flow cytometry analysis, B.L. and Y.G. performed karyotype analysis; C.W. and J.S. wrote the paper with input from W.G., L.L., F.E., H.B., T.W., and M.Y.

## Competing interests

The authors declare no competing interests.
