## [Peer review file · Communications Biology]

Reviewers' comments:

Reviewer #1 (Remarks to the Author):

This article is presenting the phased chromosome scale genome assembly and annotation of 4 lineages of *Phragmites australis*, a widespread Poaceae species. Comparative genomics has shown that hybridization and polyploidization events can contribute to the adaptation of plants in new environments. The expansion of gene families seems to play a role in invasiveness. The methods used are well described and the results seem of quality. The conclusions drawn by the authors are relevant. The paper is well structured and easy to read.

I have only few remarks:

- In the table 1, the % of N in the scaffolded assemblies are missing. I assume that the scaffolding process with Ragtag led to gaps in the scaffolds.
- In the methods, the authors speak about RNA extraction. The methods and the sequencing are not described. But in the Reference genome annotation, the authors presents RNAseq data downloaded from NCBI. Did they realize RNAseq?
- Have the authors placed the scripts used for their analysis in a github repository?
- In the Figure S1, some links are still visible between two scaffolds (around 700Mb). This is also the case between chr 7 and chr 11 (Group 7), and between Chr 19 and 20 (Group 9). Can the authors comment those contacts?

Reviewer #2 (Remarks to the Author):

Common reed is an important grass species, except for its invasive, it has been used as a traditional Chinese medicine. Authors analyzed lots of results; however, I can't find more important biological knowledges expect data driven. And the analyses are simple and the writing seems poor and unfocussed. Authors should rephrase the introduction, results and discussion.

1. The quality of English should be considerably improved in this manuscript.
2. Some typing errors: Line10 in Page 4. It deletes 'don't'. Line13 in Page 4. 'should be further investigated'.
3. Line 13-15 in Page 4. I can't understand the meaningful of common ragweed for this manuscript. Similarly, Line 8 in Page 5.
4. Line 12 in Page 6. Delete 'Six contigs were ... initial contig level assembly'.
5. The CN genome may have too many genes (73,066) annotated. In addition, I can't find gene number from other genomes in Table 1. And Table 1 should be concise.
6. Software and method used in three representative assemblies should be transferred to method and materials.
7. Chromosome number should be sort in Figure 1B, and 'other contigs' should be deleted. Figure 1E should be deleted.
8. I don't find more useful information about asexual reproduction.

Reviewer #3 (Remarks to the Author):

This paper presents a comprehensive analysis of Common reed (*Phragmites australis*), a highly invasive species, through a detailed chromosome-scale assembly and genomic investigation. The authors present a high-quality genome assembly and anchored the contigs to 24 pseudochromosomes and one B chromosome. Based on the comparative genomics analysis, they found that gene family expansions may have contributed to the invasiveness of the lineage. But the present analysis cannot answer the mechanisms of invasion, as genome-wide duplication and gene family expansion are common for plants. A few other concerns are as follows:

1. the structure of the article and the picture need to be adjusted, each picture is best corresponding to one or two chapters, do not mix.

2. The serial number of the pictures should be arranged according to the content of the article (A-B-C-D), but the figure 2F in the article is in front of the 2D; Figure 4 is even more mixed.
3. The language needs to be revised and polished throughout the whole article. Many sentences are grammatically incorrect, such as 18 lines on page 14 and 7 lines on page 15; There are also many inappropriate uses of the in and to prepositions.
4. There are many errors in the writing of details in the paper, such as 903Mbp in Abstract, there is no space between numbers and units, and Mbp is changed to Mp in the text. What does "287B USD" mean in line 8 of Introduction? There are many similar writing errors in the article, and it is recommended to revise the whole article.
5. The authors detected 13 GO processes that were specific in the invasive lineage, and how many genes were involved in those processes. The authors later verified that only 25% of gene expansion is tandem duplication, so how many genes are involved in the 13 GO processes. A quarter of the tandem genes with a dosage effect isn't a little inappropriate?
6. The Venn diagram of 3E is not labeled, and it is not known what these numbers represent. The description of the results is not shown in the figure, and are the differences in transposons due to sequencing biases?
7. Although the authors annotated many functional genes on the B chromosome, whether and how much of these genomes were expressed in tissues (root, stem, leaf, flower and seed) was unclear. Because the B chromosome is generally inert, genes are generally not expressed. If the authors found that B chromosome genes were not expressed, there was no relationship with growth or fitness.
8. The authors can increase the influence of B chromosome genes on the transcription of A and D chromosome genes, especially in the growth and proliferation. References: Effect of aneuploidy of a nonessential chromosome on gene expression in maize

Reviewer #1 (Remarks to the Author):

Comment 1: This article is presenting the phased chromosome scale genome assembly and annotation of 4 lineages of *Phragmites australis*, a widespread Poaceae species. Comparative genomics has shown that hybridization and polyploidization events can contribute to the adaptation of plants in new environments. The expansion of gene families seems to play a role in invasiveness. The methods used are well described and the results seem of quality. The conclusions drawn by the authors are relevant. The paper is well structured and easy to read.

Response: We greatly appreciate the reviewer's careful evaluation and are thankful for the positive comments on our work.

Comment 2: I have only few remarks:

- In the table 1, the % of N in the scaffolded assemblies are missing. I assume that the scaffolding process with Ragtag led to gaps in the scaffolds.

Response: Ns have indeed been added to scaffolds to fill the gaps between contigs. In the updated version of genome, we have included the proportion of Ns in the "Percentage of Ns" row in Table 1.

Comment 3: - In the methods, the authors speak about RNA extraction. The methods and the sequencing are not described. But in the Reference genome annotation, the authors presents RNAseq data downloaded from NCBI. Did they realize RNAseq?

Response: Thank you for bringing up the missing information regarding RNAseq. We initially generated one RNAseq library from the reference individual to annotate the genome assembly. This library has been uploaded to NCBI (with accession ID SRR19646717) later on, and we included the NCBI accession number for this sample in the manuscript.

For the revised genome, we generated six RNAseq libraries from three different genetic lineages (unpublished) to further improve the annotations. These have all been uploaded onto NCBI. In addition, we downloaded 13 publicly available *P. australis* RNAseq libraries from NCBI. Therefore, in total, we used 20 RNAseq libraries for genome annotation. The NCBI accession numbers for these samples are given in Supplementary Table S10.

In the revised version, we have included their description in the Materials and Methods section, line 16-21, Page 22: "To improve the gene model predictions, we incorporated 20 RNAseq libraries for genome annotation, including 13 RNAseq samples downloaded from NCBI (Supplementary Table S10). Some of these data were originally used to annotate the genome assembly of the EU invasive lineage¹. Additionally, we sequenced seven RNAseq libraries representing three different genetic lineages (CN, Med, and South Africa)."

Comment 4:- Have the authors placed the scripts used for their analysis in a github repository?

Response: Thank you for your kind suggestion. We have palced the scripts used for this manuscript in a github repository (https://github.com/smallfishcui/Phrag_Genome) and updated the information in the Data Availability Statement section. Some analyses were

done by a few simple awk commands handling results from websites, such as CoGe, thus were not written into scripts.

Comment 5: In the Figure S1, some links are still visible between two scaffolds (around 700Mb). This is also the case between chr 7 and chr 11 (Group 7), and between Chr 19 and 20 (Group 9). Can the authors comment those contacts?

Response: Thanks for pointing these out! It is possible that the links are biological, for example due to a large interchromosomal duplication, but we considered this scenario more unlikely than the alternative, assembly errors resulting from, for example, organellar genomes generating chimeric contigs during the assembly process.

In the revised MS, to check whether the chromosomes were assembled correctly, we reassembled the genome. In the revised assembly we first assembled the organellar genomes, complete circular sequences of the chloroplast and mitochondria, and then carried out nuclear genome assembly with reads that do not map to the organelles. The subsequent Hifiasm assembly was then scaffolded into chromosomes using Allhic scaffolding.

The improved procedure yielded an assembly where all chromosomes were running from telomere to telomere with no gaps. During this process also the HiC contact problems were resolved. Please refer to Supplementary Figure S2 for the visualization of the Hi-C contact map of the latest genome version.

Reviewer #2 (Remarks to the Author):

Comment 1: Common reed is an important grass species, except for its invasive, it has been used as a traditional Chinese medicine. Authors analyzed lots of results; however, I can't find more important biological knowledges expect data driven.

Response: Thank you for highlighting the medical importance of the target species, we added also its use in traditional Chinese medicine to the introduction.

We have revised the introduction and the text throughout to provide a more comprehensive overview of the biological background of *Phragmites*. We have expanded the text on the biogeographic lineages and focused the text on the key traits that distinguish invasive and non-invasive individuals. Please see the updates in Line 22-30, Page 4, and Line 4-5, Page 5.

Comment 2: And the analyses are simple and the writing seems poor and unfocussed. Authors should rephrase the introduction, results and discussion.

Response: We have now refreshed and clarified the introduction, results and discussion. We added more detail in the methods and materials section to clarify our work. The main contribution of the manuscript is to provide resources for the genome analysis of this highly invasive species:

- (i) the first high quality, chromosome-level genome assembly for an invasive plant species (and in fact one of the globally most invasive species). In the revised version we further improved the genome assembly such that the chromosomes are fully gap-free, running from telomere to telomere.

- (ii) The assemblies of other existing lineages that are also scaffolded to chromosome-level. In this revision we have further improved our assembly such that it is now a full telomere-to-telomere assembly with no gaps.

These resources will in future facilitate our understanding of the genome variation across the entire species, here we demonstrated this by investigating the evolutionary history of the species and the subgenome divergence following the polyploidy event. We found that genome evolution in terms of tandem duplications had potentially contributed to the invasiveness of *Phragmites* lineages, which provides us a hypothesis to be tested with other invasive species when more chromosome-level assemblies become available.

The genome assemblies and genome evolution were analyzed using using state of the art methods, using pipelines that were developed by our own team for the purpose of analysing polyploid species.

Comment 3:1. The quality of English should be considerably improved in this manuscript.

Response: Thanks, we have improved the English by first using AI to correct the text (chatGPT) and then asking help from a native English speaker.

Comment 4:2.. Some typing errors: Line10 in Page 4. It deletes 'don't'. Line13 in Page 4. 'should be further investigated'.

Response: Thank you for pointing this out. This is corrected and revised as suggested now.

Comment 5: 3. Line 13-15 in Page 4. I can't understand the meaningful of common ragweed for this manuscript. Similarly, Line 8 in Page 5.

Response: Artemisia is another interesting invasive plant species with a recently published genome assembly, which is why it is a landmark study. On Line 13-15, Page 4 we explained factors that may have lead to invasiveness in Artemisia. Introgression observed in Artemisia is relevant to *Phragmites* as well. We have clarified the text in the introduction by adding references to introgression found in *Phragmites* lineages into the introduction, Line 8-10, Page 5.

Line 8 in Page 5, (Line 17, Page 5 in the revised version) (“Similar to rice (*Oryza sativa*), the basal chromosome count in genus *Phragmites* is $n=12$. In *P. australis*, the euploid chromosome count varies, ranging from $3x$ to $12x$ ”) is one source of evidence that suggests that our reference assembly is from an allotetraploid species. Since the basal number in genus *Phragmites* is 12, and we identified 25 chromosomes in our reference assembly.

Comment 6:4. Line 12 in Page 6. Delete ‘Six contigs were ... initial contig level assembly’.

Response: This sentence provided evidence that the underlying assembly is highly contiguous, because six cromosomes were assembled without gaps from end to end. We have removed this sentence from the manuscript. For this revised version we further improved the genome assembly such that all chromosomes are assembled from end to end. We now provide a Supplementary Figure S3 showing the positioning of the telomeres to illustrate this point, and add the text in Line 14-16, Page 6: “Telomeres were detected at both ends of all chromosomes suggestive of complete telomere-to-telomere genome (Supplementary Figure S3). All gaps in the pseudochromosomes were filled.”

Comment 7:5. The CN genome may have too many genes (73,066) annotated. In addition, I can't find gene number from other genomes in Table 1. And Table 1 should be concise.

Response: Thank you for pointing out this. We realized that the number may have contained transposable elements which were predicted as genes by *ab initio* methods. To get more accurate genome annotations, we included data from 20 RNAseq libraries and used repeat annotation data when combining the evidence from gene predictors. This eventually yielded 42498 genes. The annotations for the other assemblies (based on illumina short reads) were obtained by transferring these annotations using liftoff and confirmed with GeMoMa.

The gene counts of each of the assemblies have been added to Table 1. The gene annotation of the EU invasive lineage comes from the original publication. We simplified Table 1 by removing subgenome information.

Comment 8:6. Software and method used in three representative assemblies should be transferred to method and materials.

Response: Thanks, we have moved this part of text to methods and materials Line 31-34 Page 21, as suggested.

Comment 9:7. Chromosome number should be sort in Figure 1B, and 'other contigs' should be deleted. Figure 1E should be deleted.

Response: We have made the adjustment accordingly and moved Figure 1E to Supplementary Figure S1.

Comment 10:8. I don't find more useful information about asexual reproduction.

Response: Asexual propagation has been suggested as one possible escape from extinction, but it incurs a high genome cost in terms of Mueller's ratchet, since in a clonally propagating species there is no recombination to break the linkage between beneficial alleles and deleterious alleles. The analyses carried out in this part of the manuscript highlight the increase of deleterious load when a species transfers to this mode of propagation.

The analysis illustrated that the invasive lineage may rely more on sexual reproduction. This strategy allows for the natural selection and recombination to aid in selecting advantageous traits and maintaining higher genetic diversity within the population. Conversely, the North American native lineage predominantly utilizes asexual propagation, leading to the accumulation of deleterious alleles and a less well-maintained population. We have now clarified the meaning and implications of this analysis in the paper, Lines 31-32, Page 18.

Reviewer #3 (Remarks to the Author):

Comment 1:This paper presents a comprehensive analysis of Common reed (*Phragmites australis*), a highly invasive species, through a detailed chromosome-scale assembly and genomic investigation.

The authors present a high-quality genome assembly and anchored the contigs to 24 pseudochromosomes and one B chromosome. Based on the comparative genomics analysis, they found that gene family expansions may have contributed to the invasiveness of the lineage.

Response: Thank you for acknowledging our efforts and accurately summarizing our key take-home messages.

Comment 2: But the present analysis cannot answer the mechanisms of invasion, as genome-wide duplication and gene family expansion are common for plants.

Response: We agree, genomics analyses alone are not enough to understand the mechanisms of invasion. However, they can produce novel hypotheses to be tested through controlled experiments.

Here, we identified expansions of gene families that contribute to increased proliferation. Notably, in non-invasive plants, these expansions appear to be more closely linked to environmental responses, this has been found independently in many other species. This distinction becomes more apparent with the improved genome annotation in our revised version, as the non-invasive lineages do not have expansions of the same gene families and the invasive lineage. A second prominent feature of the invasive lineage is that it has more tandemly duplicated genes than other genetic lineages, even when compared against our high quality reference assembly which is more contiguous and therefore has more complete gene space.

We included a sentence that states the difference to non-invasive species, supported by references to literature where tandem duplications have been analyzed, Line 6-8, Page 12: “Tandem duplications in non-invasive plant species play a role in environmental adaptations, whereas in invasive common reed there seems to be a greater emphasis on maintaining DNA integrity.”

1. Almeida-Silva, F. & Van de Peer, Y. Whole-genome duplications and the long-term evolution of gene regulatory networks in angiosperms. *Molecular biology and evolution* 40, msad141 (2023).

Comment 3: A few other concerns are as follows:

1、 the structure of the article and the picture need to be adjusted, each picture is best corresponding to one or two chapters, do not mix.

Response: Thanks, we have now adjusted the pictures to the right order and position.

Comment 4: 2、 The serial number of the pictures should be arranged according to the content of the article (A-B-C-D), but the figure 2F in the article is in front of the 2D; Figure 4 is even more mixed.

Response: We have rearranged the elements in each picture now.

Comment 5: 3、 The language needs to be revised and polished throughout the whole article. Many sentences are grammatically incorrect, such as 18 lines on page 14 and 7 lines on page 15; There are also many inappropriate uses of the in and to prepositions.

Response: Thank you for your suggestion. We have now used AI for language check and asked a native English speaker to read through the manuscript to improve the English.

Comment 6: 4、 There are many errors in the writing of details in the paper, such as 903Mbp in Abstract, there is no space between numbers and units, and Mbp is changed to Mp in the text. What does “287B USD” mean in line 8 of Introduction? There are many similar writing errors in the article, and it is recommended to revise the whole article.

Response: Thank you for your suggestion. We have corrected these typographical errors and spelled out the abbreviations.

Comment 7:5、 The authors detected 13 GO processes that were specific in the invasive lineage, and how many genes were involved in those processes. The authors later verified that only 25% of gene expansion is tandem duplication, so how many genes are involved in the 13 GO processes. A quarter of the tandem genes with a dosage effect isn't a little inappropriate?

Response: With the new genome annotation, we found 1245 out of 2928 PaEU lineage specific genes (42.5%) to result from tandem duplications. These genes comprise 9.5% of all tandem duplications in PaEU. Considering that (i) tandem duplicates are also involved in other biological processes and that (ii) the duplicates should have formed very recently, since the PaEU is a very recent lineage, this proportion is quite large.

With the new annotation we detected 131 PaEU specific genes (out of the 1245) to be enriched for the 14 GO biological processes, and 255 out of 13,114 tandemly duplicated genes to be involved in these 14 GO terms. A Fisher exact test conducted on these two gene sets indicated a significant overlap between them (95 genes overlapped, overlapping p-value= $8.5e^{-194}$).

Tandem duplications are prevalent in plants, and the proportion of gene content varies from 4.6% in *Craspedia variabilis* to 17% in *Arabidopsis*, 14% in rice, 16% in poplar, 35% in maize and 26.1% in apple. In our case, we identified 28.13% of the genes to be tandemly duplicated in the invasive lineage of *Phragmites australis*, which follows the general trend.

Comment 8:6、 The Venn diagram of 3E is not labeled, and it is not known what these numbers represent. The description of the results is not shown in the figure, and are the differences in transposons due to sequencing biases?

Response: The Venn diagram of 3E showed shared transposon types across different assemblies. The difference of transposon counts could be due to the sequencing bias, differing assembly qualities and the detection could also be affected by the software versions. We performed the analyses using the newer version of repeatmasker on our revised genome assemblies and we did not observe these differences any more. Therefore we removed the figure 3E from the manuscript.

Comment 9:7、 Although the authors annotated many functional genes on the B chromosome, whether and how much of these genomes were expressed in tissues (root, stem, leaf, flower and seed) was unclear. Because the B chromosome is generally inert, genes are generally not expressed. If the authors found that B chromosome genes were not expressed, there was no relationship with growth or fitness.

Response: This is a very interesting point. In the revised manuscript, we included more RNAseq libraries, which not only helps us improve the genome annotation but also allows us to observe the gene expression of the B chromosomes. We obtained the RNAseq data generated from Oh et al. 2022, including RNA from tissue rhizome, shoots, sprouts, and leaf, and also generated new RNAseq libraries of leaf tissues from different non-invasive genetic lineages.

The RNA-seq data were mapped to the reference genome, and the results suggested that genes on the B chromosomes are expressed in all individuals and tissues. Based on this mapping, it appears that all accessions with RNAseq data would possess the B chromosome. However, since we are uncertain about the number of copies of the B chromosome in these individuals, this could also be due to expression of homologous genes on other chromosomes. The RNAseq data showed that the expression levels vary across accessions and tissues and the numbers of alternatively spliced isoforms differ considerably, with shoots showing the highest numbers. The information is collected in Supplementary Table S10 and stated in Lines 32-34, Page 14 and Line 1-6, Page 15.

In future experiments, we aim to study B chromosome-specific gene expression by examining individuals with varying numbers of B chromosomes. We will compare the expression levels and estimate the number of B chromosomes using genome sequencing data from the same individual, or alternatively use chromosome staining methods to identify B chromosome counts. Without this baseline knowledge the RNAseq results can only be argumentative.

Comment 10:8、 The authors can increase the influence of B chromosome genes on the transcription of A and D chromosome genes, especially in the growth and proliferation. References: Effect of aneuploidy of a nonessential chromosome on gene expression in maize

Response: Thank you for the nice suggestion! We do intend to investigate the effect of B chromosome gene expression on the nuclear genes in the near future. However, we currently lack precise information on the exact number of B chromosomes in the available genome accessions, highlighting the need for additional controlled experiments. We have now added the citation and discussion on the implicated effects in conclusions section.

REVIEWERS' COMMENTS:

Reviewer #1 (Remarks to the Author):

I congratulate the authors on rewriting a large part of the manuscript. The corrections made have greatly improved quality.

The authors have taken advantage of the feedback to make improvements to their genomes and annotations.

For my part, they have responded to all my comments.

Reviewer #2 (Remarks to the Authors):

None

Reviewer #3 (Remarks to the Author):

The author has provided an explanation or analysis based on the question I mentioned above. I have no further questions with the current version.